# Erucin Exerts Cardioprotective Effects on Ischemia/Reperfusion Injury through the Modulation of mitoKATP Channels

**DOI:** 10.3390/biomedicines11123281

**Published:** 2023-12-12

**Authors:** Lorenzo Flori, Rosangela Montanaro, Eleonora Pagnotta, Luisa Ugolini, Laura Righetti, Alma Martelli, Lorenzo Di Cesare Mannelli, Carla Ghelardini, Vincenzo Brancaleone, Lara Testai, Vincenzo Calderone

**Affiliations:** 1Department of Pharmacy, University of Pisa, 56120 Pisa, Italy; lorenzo.flori@farm.unipi.it (L.F.); alma.martelli@unipi.it (A.M.); vincenzo.calderone@unipi.it (V.C.); 2Department of Science, University of Basilicata, 85100 Potenza, Italy; rosangela.montanaro@unibas.it (R.M.); vincenzo.brancaleone@unibas.it (V.B.); 3Research Centre for Cereal and Industrial Crops, CREA—Council for Agricultural Research and Economics, Via di Corticella 133, 40128 Bologna, Italy; eleonora.pagnotta@crea.gov.it (E.P.); luisa.ugolini@crea.gov.it (L.U.); laura.righetti@crea.gov.it (L.R.); 4Interdepartmental Research Center Nutrafood “Nutraceuticals and Food for Health”, University of Pisa, 56120 Pisa, Italy; 5Interdepartmental Research Centre of Ageing Biology and Pathology, University of Pisa, 56120 Pisa, Italy; 6Pharmacology and Toxicology Section, Department of Neuroscience, Psychology, Drug Research and Child Health (NEUROFARBA), University of Florence, 50139 Florence, Italy; lorenzo.mannelli@unifi.it (L.D.C.M.); carla.ghelardini@unifi.it (C.G.)

**Keywords:** erucin, isothiocyanates, mitochondrial channels, mitoK_ATP_ channels, H_2_S, H_2_S donors, 5-HD, ischemia/reperfusion

## Abstract

Modulation of mitochondrial K channels represents a pharmacological strategy to promote cardioprotective effects. Isothiocyanates emerge as molecules capable of releasing hydrogen sulfide (H_2_S), an endogenous pleiotropic gasotransmitter responsible for anti-ischemic cardioprotective effects also through the involvement of mitoK channels. Erucin (ERU) is a natural isothiocyanate resulting from the enzymatic hydrolysis of glucosinolates (GSLs) present in *Eruca sativa* Mill. seeds, an edible plant of the *Brassicaceae* family. In this experimental work, the specific involvement of mitoK_ATP_ channels in the cardioprotective effect induced by ERU was evaluated in detail. An in vivo preclinical model of acute myocardial infarction was reproduced in rats to evaluate the cardioprotective effect of ERU. Diazoxide was used as a reference compound for the modulation of potassium fluxes and 5-hydroxydecanoic acid (5HD) as a selective blocker of K_ATP_ channels. Specific investigations on isolated cardiac mitochondria were carried out to evaluate the involvement of mitoK_ATP_ channels. The results obtained showed ERU cardioprotective effects against ischemia/reperfusion (I/R) damage through the involvement of mitoK_ATP_ channels and the consequent depolarizing effect, which in turn reduced calcium entry and preserved mitochondrial integrity.

## 1. Introduction

The modulation of cardiac electrophysiological propagation depends on an orderly generation sequence of the cardiac action potential. Different internal and external ionic currents regulate the depolarization and repolarization phases through the involvement of specific membrane ion channels. Specifically, the repolarization phase involves potassium (K^+^) channels [1]. Several studies have focused on the modulation of K^+^ channels from a cardioprotective perspective in order to reproduce the extensively documented positive anti-ischemic effects due to the ischemic pre-conditioning (IPC) mechanism, a self-defense phenomenon characterized by brief periods of I/R able to increase the resistance to the injury due to a subsequent, more prolonged ischemic episode [2]. Later, another preventive phenomenon was described and named as ischemic post-conditioning (IPostC). It is characterized by brief periods of ischemia/reperfusion (I/R) at the start of reperfusion. IPC as well as IPostC have been extensively studied and numerous intracellular pathways have been identified. Among these, K^+^ channels expressed on the inner mitochondrial membrane (mitoK channels) represent the most topical and interesting pharmacological target in the cardioprotection against I/R injury [3].

The opening of mitoK channels allows the entry of K^+^ according to the electrochemical gradient. Then, the K^+^/H^+^ exchanger supports a greater exchange of K^+^ with extramitochondrial H^+^, promoting a slight mitochondrial uncoupling (5–10 mV) which reduces the production of ATP [4,5]. Furthermore, the incoming K^+^, being osmotically active, produces a slight swelling of the mitochondrial matrix, moving the inner mitochondrial matrix closer to the outer mitochondrial matrix and therefore facilitating ionic exchanges [5,6,7,8]. Furthermore, different experimental evidence highlights how the opening of mitoK channels makes isolated mitochondria more resistant to the entry of Ca^2+^ and to the triggering of apoptotic and necrotic processes due to the opening of the mitochondrial permeability transition pore (MPTP) [9,10,11,12]. Another interesting dual effect to highlight in the anti-ischemic activity exerted by the opening of mitoK channels and the cardioprotective mechanism of IPC involves reactive oxygen species (ROS). Firstly, the production of low but significant levels of ROS as a result of alkalization due to electron transport within the inner mitochondrial membrane take part, with surface receptors, in the activation of downstream protein kinases (PKCs). Then, it modulates the large ROS production and the consequent damage in the post-ischemic reperfusion phase [13].

Different types of mitoK channels, such as big conductance calcium-activated (mitoBK) and small conductance calcium-activated (mitoSK), ATP-sensitive (mitoK_ATP_), and, more recently, the voltage-gated 7.4 (mitoKv7.4) potassium channels, have been described for their own cardioprotective effects [14,15,16,17].

Of note, I/R injury, characteristic of acute myocardial infarction and the subsequent reperfusion phase represents a complex process involving several mediators. In this context, the endogenous gasotransmitter H_2_S has emerged as an important player. The inhibition of its endogenous production is associated with a higher occurrence and a lower tolerance of ischemic events and linked with an increase in infarct size on isolated hearts subjected to I/R injury [18,19,20]. Knockout mice for cystathionine-γ-lyase (CSE) enzyme, which represents the main pathway of endogenous H_2_S production in the cardiovascular system, show greater myocardial damage compared to the wild type [21]. H_2_S exerts cardioprotective effects through the regulation of a plethora of mechanisms, including the regulation of redox signals and the interaction with reactive oxygen species (ROS) associated with myocardial I/R injury [22], the modulation of anticalcifying effects [23], the cross-talk with nitric oxide (NO), and the consequent upregulation of cGMP pathway [24]. Moreover, H_2_S may positively regulate also mitoK_ATP_ and mitoBK channels, and such effect is correlated to anti-ischemic cardioprotection [25,26,27,28]. Worthy of mentioning, recent evidence demonstrates that H_2_S can modulate many intracellular pathways through the persulfidation mechanism, a post-translational modification on cysteine residues of specific protein targets [29]. On the basis of these premixes, natural and synthetic H_2_S donor compounds might act as prodrugs [30] releasing the gasotransmitter in biological environments and furthermore improving post-infarct myocardial recovery in several murine models [27,31,32].

Recently, our research group has demonstrated how the isothiocyanate moiety, characteristic of several compounds of natural origin, is capable of releasing H_2_S and exerts cardioprotective effects [27,32,33]. In this regard, erucin (ERU) is an isothiocyanate deriving from the enzymatic hydrolysis of the GSL glucoerucin, naturally occurring in the different cellular compartments of *Eruca sativa* Mill., a plant belonging to the *Brassicaceae* family. Interestingly, ERU has emerged as an interesting smart, long-lasting H_2_S donor, leading us to suppose that—at least in part—its beneficial effects are due to this gasotransmitter [30]. The *Eruca sativa* seed extract, likely through a synergistic mechanism between H_2_S, released by GSLs and the corresponding isothiocyanate ERU, and phenolic acids, highlights a positive effect in the containment of blood pressure in spontaneously hypertensive rats and a cardioprotective effect in animal model of acute myocardial infarction [34]. Furthermore, the isolated isothiocyanate ERU, in addition to the protective effects towards oxidative stress in endothelial and vascular smooth muscle cells, very recently showed cardioprotective effects in cardiomyoblast cells exposed to H_2_O_2_ and in vivo in a murine model of acute myocardial infarction. Very interestingly, Testai and colleagues first demonstrated that ERU executed protective effects through the persulfidation of the more recently discovered, mitoKv7.4 channels [35,36,37].

Considering that the most extensively studied mitoK channel is the mitoK_ATP_ one, H_2_S is recognized as an endogenous activator of this channel, and furthermore, H_2_S-donors are able to play cardioprotective effects through its stimulation; in this work, we focused on the evaluation of the involvement of mitoK_ATP_ channels in the cardioprotection showed by ERU.

## 2. Materials and Methods

### 2.1. Erucin Production

ERU (Figure 1) was produced by myrosinase-catalyzed hydrolysis of glucoerucin isolated from *Eruca sativa* Mill. defatted seed meals, as described in [33]. Briefly, the glucosinolate was extracted from *E. sativa* (Nemat) defatted seed meal in 70% ethanol and it was isolated by an ion exchange chromatography on a DEAE-Sephadex A-25 anion-exchange column (5 × 20 cm) (GE Healthcare Bioscience AB, Uppsala, Sweden) pre-conditioned with 50 mM acetate buffer, pH 4.2. After washing procedures, glucoerucin was eluted in two fractions with 0.2 M and 0.5 M K_2_SO_4_. The eluates were concentrated using a rotary evaporator and the glucosinolate was precipitated adding absolute ethanol previously cooled to −20 °C. The precipitated glucoerucin K+ salt was centrifuged and extracted again in boiling methanol to remove the excess of K_2_SO_4_. The methanolic glucosinolate solution was concentrated again to dryness before resuspension in the minimum amount of distilled H_2_O. The solution was filtered (0.2 µm Millipore, Burlington, MA, USA) and the purity of glucoerucin was further improved by gel-filtration in a XK 26/100 column packed with Sephadex G10 chromatography media (GE Healthcare Bioscience AB, Uppsala, Sweden), connected to an AKTA-FPLC System (Amersham Bioscience AB, Uppsala, Sweden). Individual fractions containing pure glucosinolate in distilled H_2_O were pooled and freeze-dried before HPLC-UV analysis. Glucoerucin analysis was performed by using a Hewlett-Packard 1100 HPLC equipped with a diode array detector and an Inertsil 5 ODS-3 column (250 × 3 mm), following the ISO 9167-2019 method [38]. The myrosinase was preventively isolated from ripe seeds of white mustard (*Sinapis alba* L.) according to Pessina et al. [39] and stored at 4 °C in sterile distilled H_2_O until use. Glucoerucin was finally hydrolyzed in the presence of in 0.1 M potassium phosphate buffer, pH 6.5, for 2 h a 37 °C. After hydrolysis, ERU was extracted in dichloromethane in a separating funnel and the collected organic phases were pooled and concentrated to dryness at 37 °C in a rotary evaporator under vacuum. The remaining dichloromethane was removed under N_2_ flow to constant weight. ERU was obtained in form of a clear oil and was finally identified by gas chromatography–mass spectrometry analysis (GC-MS) comparing the retention time and MS spectra (161 (M+,9), 145 (11), 115 (100), 85 (36), 72 (53), 61 (80)) with the purchased analytical standard ERU (Santa Cruz sc-204741) analyzed under the same chromatographic conditions and with the NIST/EPA/NIH Mass Spectral Database (NIST 11, Gaithersburg, MD, USA). GC-MS analysis was performed on a Scion 436-GC equipped with an HP-5 fused silica capillary column (30 m × 0.25 mm i.d.; 0.25 μm film thickness, J&W Scientific Inc, Folsom, California) connected to a single quadrupole mass detector (Bruker Scion SQ Premium, Bruker Daltonics, Macerata, Italy). ERU purity was calculated after analysis on an Agilent 7820A GC-FID, equipped with a Zebron XB-5MSi column (30 m × 0.25 mm i.d., 0.25 μm film thickness, Phenomenex, CA, USA), using benzyl isothiocyanate (99.8%, Sigma-Aldrich, Milan, Italy) as an internal standard and maintaining the same conditions described in Ugolini et al. [40].

### 2.2. Animal Procedures

Male Wistar rats of about 300–400 g were selected for animal tests. All the procedures involving animals were authorized by the Italian Ministry of Health (authorization number for in vivo procedures: 909/2016-PR; authorization number for ex vivo procedures: 45972/2016) and carried out in accordance with EU (EEC Directive 2010/63) and Italian (DL 4 March 2014, n.26) legislation. The animals were housed under controlled conditions with constant humidity and temperature (50% and 22 °C, respectively) and with free access to food and water and total freedom of movement. The animals were exposed to 12 h dark/light cycles.

#### 2.2.1. In Vivo Acute Myocardial Infarction

In order to evaluate the cardioprotective effect of ERU, a preclinical model of acute myocardial infarction was reproduced in rats through the ligation of the left anterior descending coronary artery (LAD). In particular, ERU and diazoxide (Merck KGaA, Darmstadt, Germany) were administered via i.p. injection 2 h before the experimental procedure at 10 mg/kg and 40 mg/kg, respectively. An equivalent volume of DMSO (1 mL/kg) corresponding to the vehicle used for the drugs was administered in control animal group. The Sham group was subjected to the whole surgical procedure without LAD, in order to evaluate the impact of the experimental procedure itself on cardiac damage. Furthermore, in order to evaluate the contribution of IPC on cardioprotection against I/R damage, a group of animals was subjected to two cycles of 5 min of ischemia followed by 10 min of reperfusion before inducing 30 min of coronary occlusion. Selective blocker of mitoK_ATP_ channels (5-hydroxidecanoic acid, 5-HD, Merck KGaA, Darmstadt, Germany) was injected i.p. at the dose of 10 mg/kg 20 min before the injection of ERU (Figure 2).

To describe the procedure in detail, the animals were previously anesthetized by i.p. injection of sodium thiopental (MSD, Milano, Italy) (70 mg/kg) to keep the animal under deep sedation. Subsequently, a tracheotomy was performed to allow assisted breathing (1.2 mL of air/100 g of body weight, with a frequency of 70 breaths/min) (RWD Life Science Co., Sugar Land, TX, USA) and performed to exclude lung collapse and therefore reduction in respiratory function. Then, the heart was exposed and LAD was applied with a 6.0 surgical wire (Johnson & Johnson, New Brunswick, NJ, USA). Finally, the rib cage was sutured using 3.0 surgical thread (Johnson & Johnson, New Brunswick, NJ, USA). The experimental protocol was characterized by 30 min of LAD ischemia and 120 min of reperfusion. At the end of the two hours of reperfusion, intracardiac blood sampling was performed. The collected blood was transferred to a tube containing no coagulation inhibitors, it was allowed to clot for 30 min, and then it was centrifuged at 3200× *g* for 10 min using a Sigma 3-18KS centrifuge (Osterode am Harz, Germany). At the end of the centrifugation, the serum was collected and subsequently frozen at −80 °C until it was used for the analysis of troponin I levels. Furthermore, at the end of the reperfusion, the heart was removed and set up on the Langendorff apparatus (ADInstruments NZ Limited, Dunedin, New Zealand) and perfused for approximately 5 min, at constant pressure, with Krebs saline solution (NaHCO_3_ 25 mM, NaCl 118 mM, KCl 4.8 mM, MgSO_4_ 1.2 mM, CaCl_2_ × H_2_O 1.6 mM, KH_2_PO_4_ 1.2 mM, Glucose 11.5 mM, pH 7.4) (Merck KGaA, Darmstadt, Germany) previously carbogenated and thermostatted at 37 °C in order to clean the organ and coronary vessels from remaining blood. Subsequently, the heart was weighed, deprived of both atria and right ventricle, and then frozen for 30–40 min at −20 °C, after which it was cut into slices approximately 1–2 mm thick, excluding the section above the occlusion point, for the morphological analysis of necrotic areas. The procedure used for the evaluation of the extension of the ischemic areas involved 2,3,5-triphenyltetrazolium chloride (TTC) (Merck KGaA, Darmstadt, Germany). The slices obtained were immersed in TTC solution (10 mg/mL; in 1xPBS at pH 7.4 and 37 °C) and incubated at 37 °C for 20 min. This colorimetric assay allowed us to identify vital areas from ischemic necrotic ones. Finally, the ventricle sections were fixed with 10% formaldehyde (Merck KGaA, Darmstadt, Germany) overnight. After 12 h, heart sections were treated with H_2_O_2_ (Merck KGaA, Darmstadt, Germany) for 10 min to amplify the ischemic areas and photographed to perform the planimetric analysis with the GIMP 2.0 software.

#### 2.2.2. Evaluation of Systemic Troponin I Levels

Troponin I levels were measured in serum obtained from whole blood and collected at the end of each experiment through an intracardiac sampling before the heart removal and subsequent morphometric analysis. An ELISA commercial kit (Alpha Diagnostic, San Antonio, TX, USA) was used for quantitative analysis of troponin I levels following the manufacturer’s directions.

### 2.3. Ex Vivo Procedure

Sodium thiopental (100 mg/kg) was used to sacrifice male Wistar rats. An Ultra-Turrax homogenizer (IKA1-Werke GmbH & Co., Staufen, Germany) was used to homogenize the heart in 20 mL of ice-cold isolation buffer (IB, containing 1 mM EGTA, 5 mM Tris, 250 mM sucrose; pH 7.4). The supernatant obtained after centrifugation at 1075× *g* for 3 min at 4 °C was stored on ice. The pellet was resuspended and again centrifuged under the same conditions described above to maximize the mitochondrial extraction. After merging the two supernatants, another centrifuge at 11,950× *g* for 10 min at 4 °C was carried out. The mitochondrial fraction was resuspended in the IB and centrifuged again. A third centrifugation was performed under the same experimental conditions on the obtained pellet using an EGTA-free IB. A minimal volume (about 400 μL) of the EGTA-free IB was used to resuspend the final mitochondrial pellet, which was then stored on ice for at maximum 2 h. A Bradford assay was used for the determination of mitochondrial protein concentration.

#### 2.3.1. Evaluation of Mitochondrial Membrane Potential

Mini-electrodes sensitive to tetraphenylphosphonium (TPP^+^) coupled with a reference electrode (WPI-World Precision Instruments, Sarasota, FL, USA) were used to determine the mitochondrial membrane potential (Δ*ψ*) with a potentiometric approach. Biopac Inc. (Goleta, CA, USA) was used for data acquisition. Mitochondria (1 mg of protein/mL) were suspended under gentle and constant shaking in the incubation medium (IM, containing 10 mM TPP^+^Cl^−^, 10 mM Hepes, 120 mM KCl, 45 mM K_2_HPO_4_, 2 mM MgCl_2_, 10 mM succinic acid, 1 mM EGTA; pH 7.4). The membrane potential value was calculated by a modified Nerst equation, as previously described [41]:$$\Delta \psi =60\times log\frac{V_{0}.\frac{{\left[{\mathrm{TPP}}^{+}\right]}_{0}}{{\left[{\mathrm{TPP}}^{+}\right]}_{t}}-V_{t}-K_{0}P}{V_{m}P+K_{i}P}$$

A mitochondrial membrane potential >−170 mV was considered not in line with sufficient energization and was then discarded.

Changes in Δ*ψ* were monitored after ERU addition to the mitochondrial suspension at increasing concentrations (1–100 μM). To evaluate the involvement of mitoK_ATP_ channels on ERU-induced changes of Δ*ψ*, ATP (200 μM) was pre-incubated in the mitochondrial suspension 5 min before to the addition of ERU. The effects of the corresponding vehicle (DMSO 0.1%) were also evaluated.

#### 2.3.2. Modulation of K^+^ Flow through the Involvement of mitoK Channels

A fluorimetric approach was used to evaluate the ability of ERU to activate potassium channels using the thallium ion (Tl^+^) as a potassium-mimetic agent [42]. Isolated cardiac mitochondria were incubated for 10 min with loading buffer (LB, containing a Tl^+^-sensitive fluorescent probe) in order to allow the storage into the mitochondrial matrix. The excess of the Tl^+^-sensitive probe was removed from mitochondria with double centrifugation at 11,950× *g* for 10 min at 4 °C. The pellet was resuspended in 400 µL of IB and stored on ice until the protein assay using Bradford reagents. Prior to testing, to obtain 0.5 mg of mitochondrial protein/mL, the mitochondrial suspension was further diluted in K^+^-free incubation medium (IM) (containing 240 mM mannitol, 5 mM Na_2_HPO_4_, 10 mM Hepes, 10 mM succinic acid, 2 mM MgSO_4_, 200 μM ATP, pH 7.4). The mitochondrial suspension was placed in a 96-well plate and added with diazoxide (100 µM), ERU (30 µM) or their vehicle (DMSO 1%). To evaluate the specific involvement of mitoK_ATP_ channels, the specific inhibitor, 5HD (100 µM) was incubated 2 min before the addition of ERU. After 2 min, an aqueous solution of Tl_2_SO_4_ was added to each well. The increase in fluorescence (due to the entry of Tl^+^ into the matrix, through the potassium channels) was monitored (ex = 488 nm, em = 525 nm) for 120 s with an EnSpire multiplate reader; then, the area under curve (AUC) was calculated for different treatments and the AUC obtained with vehicle was subtracted (Appendix A).

### 2.4. Data Analysis

The ischemic area was analyzed as a percentage of the total left ventricular area (Ai/A_LV_). Circulating troponin I levels measured in serum were reported as pg/mL. Mitochondrial membrane potential was evaluated in relation to changes in mV from baseline levels both in the presence of vehicle and increasing ERU concentrations with or without pre-incubation with the physiological blocker of mitoK_ATP_ channels, ATP. The activation of the mitoK_ATP_ channels analyzed with the fluorimetric method was expressed as a percentage with respect to diazoxide; an increased percentage indicated the decrease in the Tl^+^ ion into the IM and an increased entry into the mitochondrial matrix. Each result was obtained from the heart of at least 5 different animals for each kind of treatment. Data were statistically analyzed by one-way ANOVA followed by Bonferroni post-test (software: GraphPadPrism 8.0). *p* values < 0.05 were considered indicative of statistically significant differences.

## 3. Results

### 3.1. Involvement of mitoK_ATP_ Channels in the Cardioprotective Effect of ERU—Morphometric Analysis

The acute myocardial infarction procedure, with 30 min of coronary occlusion and 120 min of reperfusion, produced a marked ischemic area in the hearts of the untreated rats (Vehicle) (38.5 ± 1.5%). The induction of the IPC mechanism confirmed, in agreement with the literature, how this process is able to “preserve” the heart for a subsequent ischemic event, making it capable of reacting promptly, as shown by the significant reduction in the ischemic area (18.5 ± 4.1%). In parallel, the use of diazoxide, a selective activator of mitoK_ATP_ channels, also confirms that the stimulation of potassium flows can be considered a key mechanism in anti-ischemic cardioprotection. In this case, the administration of the well-known mitoK_ATP_ opener diazoxide at a dose of 40 mg/kg 2 h before the ischemic event showed a significant reduction in the ischemic area (19.7 ± 1.9%) when compared with the vehicle. This result is perfectly in agreement with the literature and further confirms the role of these mitochondrial channels in the triggering of IPC. ERU, administered at a dose of 10 mg/kg, in this experimental model, induced a significant containment of the ischemic area (22.8 ± 1.7%) comparable to those observed for the IPC group and the diazoxide one. Pre-treatment with 5HD (5 mg/kg), a selective inhibitor of mitoK_ATP_ channels, was able to significantly inhibit ERU-induced cardioprotection (34.2 ± 1.7%) (Figure 3).

### 3.2. Involvement of mitoK_ATP_ Channels in the Cardioprotective Effect of ERU—Troponin I Release

Measurement of serum troponin I levels represents a well-known index of cardiac damage. The marked ischemic area observed in the vehicle group was accompanied by doubled serum troponin I levels if compared to the sham group (256.6 ± 17.7 pg/mL vs. 127.2 ± 17.7 pg/mL). On the other hand, serum troponin I was significantly reduced in the group of rats subjected to the IPC procedure (128.5 ± 28.5 pg/mL) compared to the control group treated with the vehicle alone. Again, diazoxide injection (40 mg/kg) exerted a significant reduction in serum troponin I levels (134.7 ± 25.4 pg/mL), thus confirming the cardioprotective effect observed above on the containment of ischemic area. The administration of ERU at a dose of 10 mg/kg contained serum troponin I levels (155.4 ± 17.6 pg/mL) which were comparable to the IPC and diazoxide groups. Troponin I levels raised again, along with the increase in ischemic area, with the pre-administration of 5HD (277.2 ± 39.8 pg/mL) (Figure 4).

### 3.3. Involvement of mitoK_ATP_ Channels in the Effect of ERU: Evaluation of the Mitochondrial Membrane Potential

Physiological levels of mitochondrial membrane potential in a fully functional organelle are around −180 mV. Cumulative additions of ERU in the suspension of isolated cardiac mitochondria produced a concentration-dependent mild-depolarization that reached the maximum effect at a concentration of 100 μM. This depolarizing effect was completely inhibited by preincubation of the physiological inhibitor of K_ATP_ channels, ATP (at a concentration of 200 μM) (Figure 5).

### 3.4. Involvement of mitoK_ATP_ Channels in the Effect of ERU: Evaluation of the Mitochondrial Potassium Flows

ERU 30 μM was the concentration selected for the evaluation of potassium flow. ERU promoted the opening of mitoK channels producing a fluorescence emission of 61.0 ± 9.0% compared to diazoxide 100 μM, which was used as a reference compound. Preincubation with 5HD significantly inhibited the ability of ERU to permit potassium currents across the inner mitochondrial membrane and produced a remarkable reduction in fluorescence emission vs. ERU alone (10.8 ± 4.4%) (Figure 6).

## 4. Discussion

Cardiac pathologies, whether resulting from direct causes affecting the cardiovascular system or as comorbidities resulting from metabolic pathologies or aging, represent a core challenge [43].

The acute myocardial infarction, responsible for countless deaths every year, supposes as a primary and essential objective the recovery of correct blood flow. Indeed, the restoration of the correct condition to minimize irreversible tissue damage leads to the oxidative processes responsible for the functional worsening and the triggering of the apoptotic death processes that characterize the reperfusion phase following the ischemic event [3,44].

In this regard, the discovery of the endogenous production of the gasotransmitter H_2_S and its positive cardiac effects paved the way to innovative approaches to the management of myocardial I/R damage [45].

Indeed, a considerable amount of preclinical evidence underlines the positive impact of H_2_S in myocardial infarction on I/R injury and on post-ischemic remodeling. Indeed, H_2_S is able to promote a protective effect similar to those observed in the IPC and IPostC [46,47]. Interestingly, the exogenous administration of H_2_S promotes protective effects against I/R injury also in senescent cardiac cells (H9c2) and in isolated hearts from aged rats [48,49]. Furthermore, administration of the sulfur compound diallyl disulfide (DADS) from the *Alliaceae* family reduced the infarct size and preserved contractile function after I/R injury by restoring cardiac H_2_S levels [50]. Consistently, preclinical investigations highlighted that, in addition to the acute effects of a single administration of H_2_S donors in the containment of the infarct size, repeated administrations promoted significant improvements in cardiac function and post-ischemic cardiac remodeling [51].

Among the different H_2_S donors, the isothiocyanate moiety represents a viable strategy to unleash the positive effects of H_2_S. Indeed, we demonstrated that both synthetic and natural isothiocyanates slowly release H_2_S in a thiol-dependent manner [29,30]. The synthetic H_2_S donors 4-carboxyphenyl-isothiocyanate and 3-pyridyl-isothiocyanate significantly protected cardiac tissue from ischemic injury and promoted recovery of post-ischemic tissue function [27,32]. ERU is a natural isothiocyanate derived from *Eruca sativa* Mill., belonging to the *Brassicaceae* family, and is able to release H_2_S slowly and over an extended period, with a thiol-dependent kinetics, both in cell-free and cellular models [30]. Specifically, ERU released significant levels of H_2_S in vascular smooth muscle and endothelial cells and promoted both time- and concentration-dependent H_2_S release in murine cardiomyoblasts [35,36,37]. Recently, our experimental evidence has demonstrated how ERU mitigated oxidative damage in cardiac cells (H9c2) in a concentration-dependent manner. Furthermore, ERU exhibited a cardioprotective effect in an in vivo model of I/R injury through direct persulfidation of mitoKv7.4 channels, unequivocally confirming the link between the cardioprotective effects of ERU and the H_2_S released from the isothiocyanate moiety [33]. Moreover, ERU was found to inhibit the tumor growth in human triple-negative breast cancer cells [52] as well as in a breast cancer cell xenograft mouse model; interestingly, in this case a regulation of mitochondrial fission/fusion emerges, in particular with the translocation of cofilin and Drp1 and consequent cell apoptosis [53].

As concerns the putative mechanisms underlying these protective effects, it is well-known that H_2_S may modulate a plethora of intracellular targets by persulfidation of the cysteine residues of specific proteins. In the context of anti-ischemic cardioprotection, undoubtedly the most investigated, by us and other research groups, has been the mitoK_ATP_ channel [27,28,54]. Furthermore, it is well known that a peculiarity of this gasotransmitter is the capability to exert its effects through the contemporary modulation of many intracellular targets; therefore, we decided to explore the contribution of the well-known mitoK_ATP_ channels in the cardioprotection of ERU.

Acute myocardial infarction damage was reproduced in rats through a preclinical model characterized by 30 min of left descending coronary artery occlusion followed by 120 min of reperfusion. The experimental model was used to reproduce, from a translational perspective, both acute myocardial damage due to coronary occlusion and post-ischemic reperfusion damage. According to the literature, the experimental protocol used produced an extensive ischemic area and a significant systemic release of troponin I. As regards the pharmacological treatments, we decided to carry out an intraperitoneal pre-administration before the infarct event in order to speculate on the contribution of ERU, or at least in part H_2_S release, on the IPC phenomenon. ERU 10 mg/kg was able to significantly contain both the extension of the ischemic area in the left ventricle and the serum levels of troponin I. It is noteworthy to underline that ERU has been shown to promote cardioprotective effects comparable to the IPC mechanism and to those obtained through the modulation of potassium flows. Conversely, 5HD, a specific inhibitor of mitoK_ATP_ channels, almost completely abolished the cardioprotective effect exerted by ERU, suggesting a link between ERU and mitoK_ATP_ channels. Of note, in the intracellular network involved in the cardiac protection mediated by the activation of mitoK channels, a key role is played by reactive oxidative species (ROS) production [55]. Although our previous experiments demonstrated the capability of ERU to reduce the injury induced by pro-oxidant environments [33] and the lack of an evaluation of ROS levels may represent a limitation.

In order to deeply characterize the interaction between ERU and mitoK_ATP_ channels, we chose an experimental model that minimized interactions with other undesired biological targets and decided to isolate mitochondria from heart tissue, on which we carried out functional studies to highlight the involvement of these channels.

Functional mitochondria in optimal physiological conditions have a membrane potential (ΔΨ) around −180 mV. Excessive hyperpolarization or depolarization due to drastic electrochemical imbalances, a characteristic condition of I/R, can damage mitochondria and irreversibly compromise their integrity and functionality. On the other hand, a slight membrane depolarization of approximately 20–25 mV provides greater tolerance to I/R damage. In this context, ERU produced a mild membrane depolarization in a concentration-dependent manner, reaching at the highest concentration tested (100 μM) a change in the basal value of about 20 mV. Interestingly, this effect, compatible with greater mitochondrial reactivity, was abolished by pre-incubation with 5HD, strengthening the idea of the involvement of mitoK_ATP_ channels in the cardioprotective effect of ERU.

Finally, the involvement of mitoK_ATP_ channels was confirmed through the use of a probe sensitive to Tl^+^ (used as a mimetic K^+^ ion). ERU was added to the suspension of isolated cardiac mitochondria at a concentration of 30 μM, a concentration that in previous experiments inhibited mitochondrial calcium uptake by approximately 60%. Under these experimental conditions, ERU led to the development of a fluorescence of approximately half of the diazoxide, used as reference agent in the modulation of K^+^ currents. The almost complete abolition of the effect of ERU on K^+^ currents following 5HD pre-incubation clearly confirmed the involvement of mitoK_ATP_ channels in the mechanism of action of ERU.

However, K_ATP_ channels have also been found in other cellular comportments beyond the inner mitochondrial membrane, for instance on the cellular membrane, in nuclear and endoplasmatic and lysosomal compartments, and involved in the cell protection against several type of injuries [56,57]; therefore, it is not possible to exclude that following a systemic administration of a K_ATP_ activator, the opening of K_ATP_ in other intracellular compartments contributes to the protection.

Taken together, the results obtained showed how ERU exerted cardioprotective effects against I/R damage through the modulation of K^+^ currents and, in particular, through the involvement of mitoK_ATP_ channels. Furthermore, analyses conducted on isolated cardiac mitochondria confirmed how ERU, through the modulation of mitoK_ATP_ channels, promoted a depolarizing effect, which in turn reduced calcium entry and preserved mitochondrial integrity.

## 5. Conclusions

The role of mitoK_ATP_ channels has been widely demonstrated for explaining the cardioprotective effects of H_2_S and of many H_2_S donors. However, we previously reported the involvement of another type of mitoK channel, the mitoKv7.4 one, when the cardioprotective profile of ERU, an H_2_S donor, was investigated. Therefore, to our knowledge, this is the first study in which the link between ERU and activation of heart mitoK_ATP_ channels has been demonstrated.

However, we cannot exclude that the contribution of ERU is due to a direct activation of the mitoK_ATP_ channels, in addition to the release of H_2_S; in the future, an accurate analysis by an in silico approach will be carried out. Nevertheless, at the moment, only the regulatory subunit (SUR) has been crystallized and only early experiments have been carried out [58]. Therefore, based on our results, ERU protects the myocardium against the I/R injury through the activation of both mitoK channels, mitoKv7.4, as well as mitoK_ATP_ channels. Moreover, both seem to be essential for cardiac protection since the inhibition of one almost completely abrogated the reduction in injury at the tissue level and the potassium flows and the depolarization of membrane potential at the mitochondrial level. Therefore, an unresolved aspect and a challenging topic is the speculation of the possible cross-talk between these two mitoK channels. Certainly, further research is needed to address at the understanding the interplay between the two channels in a physio-pathological scenario.

## Figures and Tables

**Figure 1 biomedicines-11-03281-f001:**
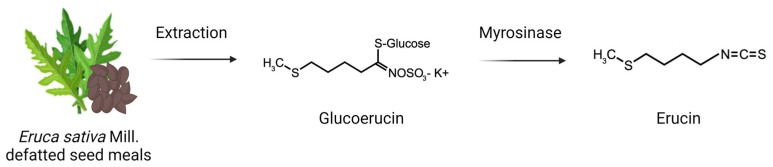
Chemical structure of ERU and graphic summary of its extraction.

**Figure 2 biomedicines-11-03281-f002:**
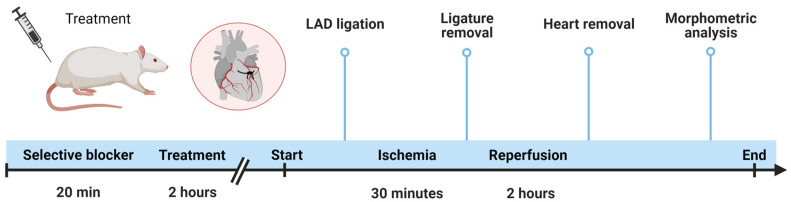
Schematic representation of the in vivo experimental protocol of acute myocardial infarct.

**Figure 3 biomedicines-11-03281-f003:**
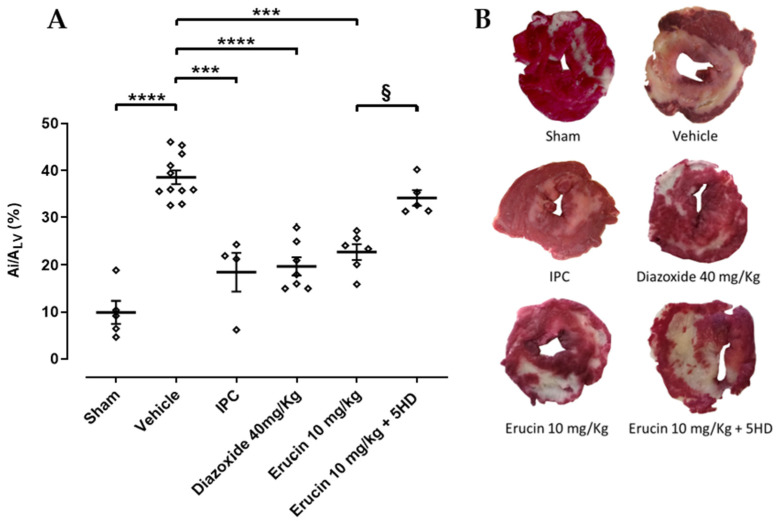
ERU-induced cardioprotection in acute myocardial infarct. (**A**) Ischemic area (A_i_) in the different groups of treatment compared to the whole left ventricle (A_LV_). * Significant difference vs. vehicle (*** *p* < 0.001; **** *p* < 0.0001). § Significant difference vs. ERU group (*p* < 0.05). Data are expressed as mean ± SEM. Number of animals was at least 5. (**B**) Exemplificative images of hearts from each different treatments after TTC staining.

**Figure 4 biomedicines-11-03281-f004:**
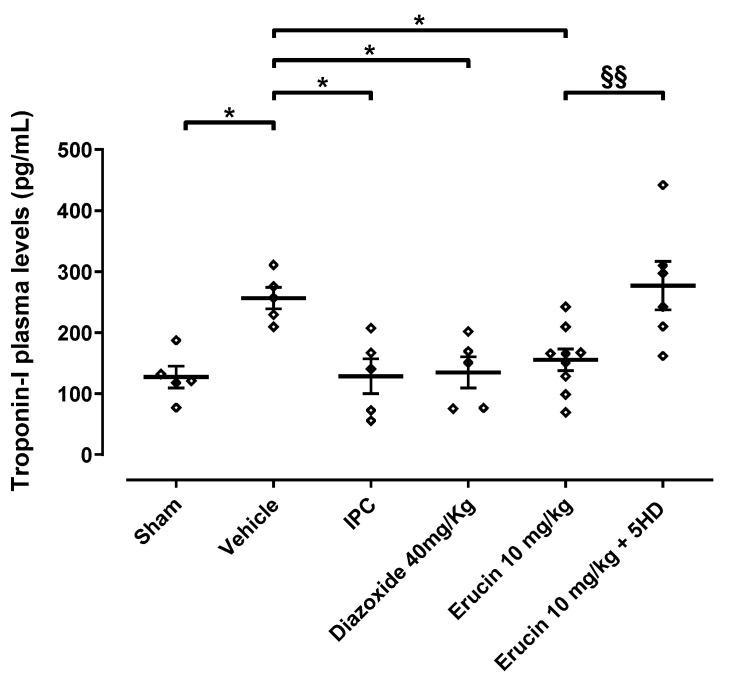
ERU-induced cardioprotection in acute myocardial infarct. Serum levels of cardiac troponin I (pg/mL) are displayed in this plot. * Significant difference vs. vehicle (* *p* < 0.05). § Significant difference vs. ERU (^§§^ *p* < 0.01). Data are expressed as mean ± SEM. Number of animals was at least 5.

**Figure 5 biomedicines-11-03281-f005:**
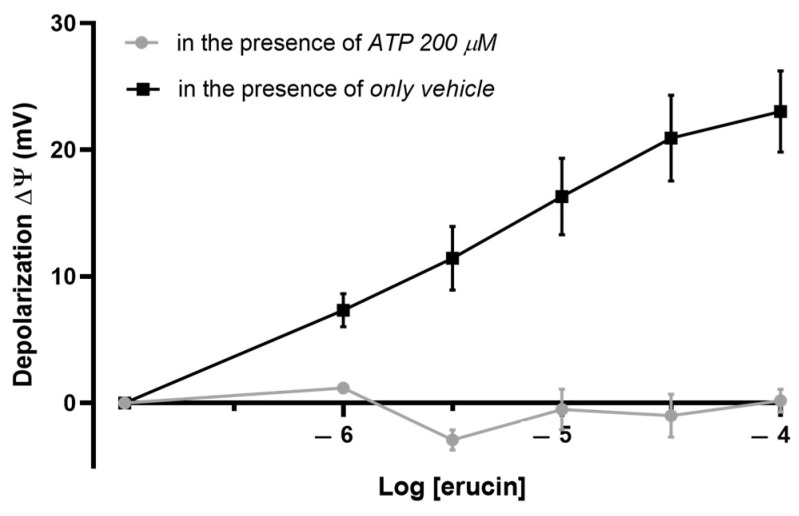
ERU effects on mitochondrial membrane potential (ΔΨ). The effects on ΔΨ (mV) of increasing and cumulative concentration of ERU are displayed in these curves in the absence or in the presence of ATP (200 μM). Data are expressed as mean ± SEM. Number of animals was at least 5.

**Figure 6 biomedicines-11-03281-f006:**
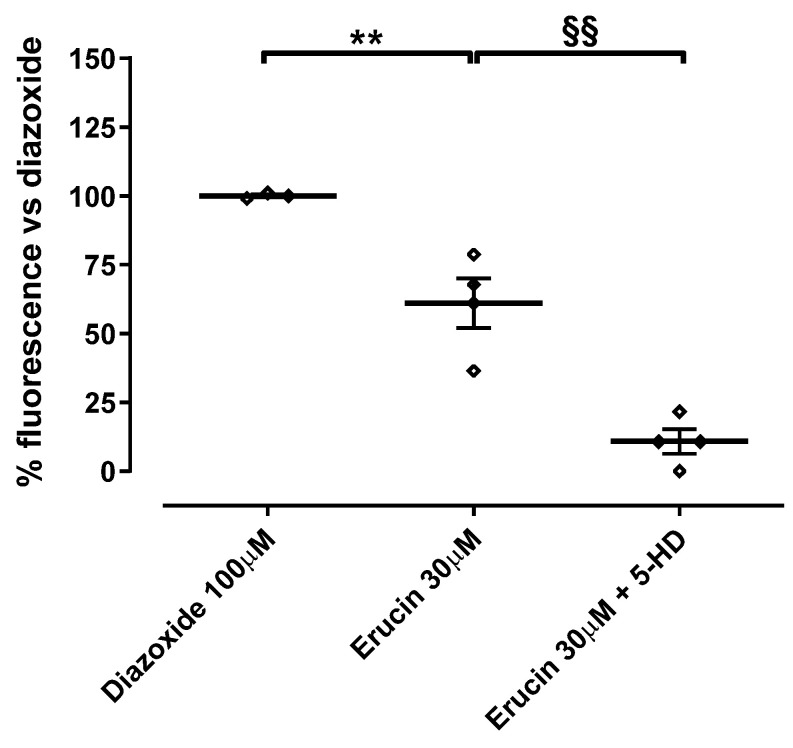
ERU effects on mitochondrial potassium flows. The plot represents the increase in the fluorescence (expressed as a % of diazoxide 100 μM) produced by a Tl^+^-sensitive probe accumulated in the mitochondrial matrix after the addition of Tl^+^ ions to a suspension in the presence of diazoxide (100 μM), ERU (30 μM), ERU plus 5HD (100 μM). * Significant difference vs. diazoxide (** *p* < 0.01). § indicates significant difference vs. ERU (^§§^ *p* < 0.01). Data are expressed as mean ± SEM. Number of animals was at least 5.

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
