# Peer review of "Erucin Exerts Cardioprotective Effects on Ischemia/Reperfusion Injury through the Modulation of mitoKATP Channels"

_biomedicines, 2023, doi:10.3390/biomedicines11123281_

Round 1
Reviewer 1 Report
Comments and Suggestions for Authors
This is a fairly high-quality study demonstrating the cardioprotective effect of natural isothiocyanate erucin associated with activation of mitochondrial potassium channels. I have several questions and recommendations.
Introduction:
The authors rather superficially indicate the cardioprotective role of mitochondrial potassium channels. It must be clearly stated that activation of potassium channels has a mild uncoupling effect on mitochondria, reducing the membrane potential and, accordingly, the intensity of ROS production in mitochondria, which prevents the development of oxidative stress. Moreover, properly functioning potassium channels maintain mitochondrial ultrastructure.
Methods:
2.1. For the convenience of the reader, it is necessary to briefly describe the procedure for producing erucin.
The experimental design involves assessing the preventive effect of erucin, but will it be effective after the development of myocardial infarction? In this case, it can be considered as an effective treatment.
Why did the authors use erucin at the concentration indicated (10 mg/kg)?
What concentration of DMSO was used as a solvent? This is important to point out.
Results and discussion:
Provide representative images of 2,3,5-triphenyltetrazolium chloride-stained myocardial samples (supplementary to Figure 1).
Provide original curves reflecting potassium transport (Fig. 4).
The therapeutic effect of potassium channel activators is due to a decrease in the level of ROS and markers of oxidative stress. The authors need to provide data on the level of these markers (for example, MDA and others), which will logically build a chain of events in the body leading to cardioprotection.
The mechanisms of regulation of mitochondrial potassium channels and potassium channels of other organelles and the cytoplasmic membrane are often similar. The authors should discuss the possible therapeutic impact of potassium channel activation in other cellular compartments.
It is known that erucin is also able to influence the processes of mitochondrial fission and fusion, which may also have a therapeutic effect on cells. The authors should discuss this possibility.
Author Response
Referee 1 This is a fairly high-quality study demonstrating the cardioprotective effect of natural isothiocyanate erucin associated with activation of mitochondrial potassium channels. I have several questions and recommendations.
Introduction:
The authors rather superficially indicate the cardioprotective role of mitochondrial potassium channels. It must be clearly stated that activation of potassium channels has a mild uncoupling effect on mitochondria, reducing the membrane potential and, accordingly, the intensity of ROS production in mitochondria, which prevents the development of oxidative stress. Moreover, properly functioning potassium channels maintain mitochondrial ultrastructure.
REPLY: we changed the introduction in order to better focus on the activation of mitoK channels.
Methods:
2.1. For the convenience of the reader, it is necessary to briefly describe the procedure for producing erucin.
REPLY: we inserted a section on the extraction of erucin from Eruca sativa Mill. seeds.
The experimental design involves assessing the preventive effect of erucin, but will it be effective after the development of myocardial infarction? In this case, it can be considered as an effective treatment.
REPLY: Thank you for the suggestion, we agree with you. Indeed, in clinical a protective agent will be administered after an ischemic event and not before; however, in the preclinical studies often a preventive approach is evaluated, in order to explore the intracellular mechanisms involved in the “pharmacological preconditioning” and point out the potential targets to plan an effective drug discovery. Certainly, a future evaluation will be focused on the treatment with ERU at the start of reperfusion.
Why did the authors use erucin at the concentration indicated (10 mg/kg)?
REPLY: preliminary experiments, carried out with Eruca sativa extract and with erucin pointed out this dose effective to obtain a cardioprotection as well as antihypertensive effect (Testai et al., Biochem Pharmacol. 2023 doi:10.1016/j.bcp.2023.115728. Testai et al., Phytother Res. 2022 doi: 10.1002/ptr.7479. Martelli et al., Br J Pharmacol. 2020 doi: 10.1111/bph.14645).
What concentration of DMSO was used as a solvent? This is important to point out.
REPLY: Thank you, DMSO was used in vivo at the dose equal to 1mL/Kg. We apologize and inserted it in the text.
Results and discussion:
Provide representative images of 2,3,5-triphenyltetrazolium chloride-stained myocardial samples (supplementary to Figure 1).
REPLY: we inserted exemplificative images for each myocardial sample and we propose to insert it as Figure 3B along the text.
Provide original curves reflecting potassium transport (Fig. 4).
REPLY: we inserted in the supplementary section the original curve deriving from a thallium assay (it is also shown below). Moreover, we inserted some details in the Materials and Methods section, to clearer explain the obtainment of the final graphic. Indeed, usually we monitored the increase in fluorescence (due to the entry of Tl+ into the matrix, through the potassium channels) for 120 seconds by EnSpire multi-plate reader; then the area under curve (AUC) was calculated for different treatments and subtracted the AUC obtained with vehicle. Finally, we considered the effect of diazoxide (positive control for the activation of mitoKATP channel) equal to 100%, and the florescence emitted after treatment with erucin or erucin+5HD was expressed as percentage of diazoxide.
The therapeutic effect of potassium channel activators is due to a decrease in the level of ROS and markers of oxidative stress. The authors need to provide data on the level of these markers (for example, MDA and others), which will logically build a chain of events in the body leading to cardioprotection.
REPLY: it would be very interesting, thank you for suggestion. Unfortunately, in this moment we are not able to carry out these evaluations. Indeed, we would need of cardiac tissue, that we used for the other analysis; therefore, we should carry out de novo experiments and our ministerial authorization is expired, then the times would be very long. We hope that this point is not mandatory, considering that we furnished two consistent markers of myocardial injury (the ischemic area and the serum levels of troponin).
The mechanisms of regulation of mitochondrial potassium channels and potassium channels of other organelles and the cytoplasmic membrane are often similar. The authors should discuss the possible therapeutic impact of potassium channel activation in other cellular compartments.
REPLY: thank you, we inserted a sentence in the discussion the suggestion.
It is known that erucin is also able to influence the processes of mitochondrial fission and fusion, which may also have a therapeutic effect on cells. The authors should discuss this possibility.
REPLY: thank you, we inserted in the discussion the suggestion.
Reviewer 2 Report
Comments and Suggestions for Authors
This paper studied the cardioprotective effect of Erucin against ischemia/reperfusion injury and the possibility of activating the mKATP ion channel by Erucin as a protective mechanism.
The authors demonstrated the ex vivo cardioprotective effect of Erucin and the indirect mKATP channel activation effect by Erucin using inhibitors and activators of the KATP channel. Some corrections and supplementary opinions regarding this paper are as follows.
Major
1. Please add a summary figure of the ex vivo experiment conditions to improve readers’ understanding of the experiment.
2. Please provide images of myocardial damage in an actual heart to make the results clear.
3. Present the structure and chemical explanation of Erucin in the form of figures, and suggest the possibility of direct binding of Erucin and mKATP channel through simulation, etc.
4. Simulate in vitro I/R injury using isolated single cardiomyocytes, and show through images whether the cardiomyocyte mitochondrial membrane potential is maintained and ROS generation is suppressed during reperfusion by Erucin.
5. Since the information related to Erucin as an H2S donor was not shown in this experiment, please delete it from the abstract and briefly describe it in the introduction.
Comments on the Quality of English Language.
Author Response
Referee 2 The authors demonstrated the ex vivo cardioprotective effect of Erucin and the indirect mKATP channel activation effect by Erucin using inhibitors and activators of the KATP channel. Some corrections and supplementary opinions regarding this paper are as follows.
Major
- Please add a summary figure of the ex vivo experiment conditions to improve readers’ understanding of the experiment.
REPLY: Thank you for the suggestion, we inserted a schematic summary of experimental conditions.
- Please provide images of myocardial damage in an actual heart to make the results clear.
REPLY: Thank you for suggestion, we inserted exemplificative images derived from TTC-treated hearts (Figure 3B).
- Present the structure and chemical explanation of Erucin in the form of figures, and suggest the possibility of direct binding of Erucin and mKATP channel through simulation, etc.
REPLY: we inserted an image of chemical structure of erucin (Figure 1). As regard the second part of the sentence, unfortunately, we are not able to establish if erucin may directly bind the mitoKATP channels; conversely, we suppose that erucin, through the release of H2S, can activate the mitoKATP channels. In fact, erucin is well-known for its H2S donor property (Citi et al., 2014 Planta Med 10.1055/s-0034-1368591; Testai et al., 2023 Biochem Pharmacol doi:10.1016/j.bcp.2023.115728; Martelli et al., 2020 Br. J. Pharmacol doi:10.1111/bph.14645).
On the other hand, while K+ currents through mitoKATP have been discovered almost 30 years ago, its molecular composition remained almost unknown until a recent study of Paggio et al., 2019 Nature doi:10.1038/s41586-019-1498-3. However, the molecular structure of the pore-forming channel is still not resolved. At our knowledge, only the regulatory subunit (mitoSUR) has been very recently crystallized (Palacio et al., 2023 doi: 0.1016/j.cbi.2023.110560); but the unique paper is focused on the physiological regulation by ATP and GTP. Therefore, at the moment no activator has been tested on this model. We inserted in the discussion, as future objective of the current study.
- Simulate in vitro I/R injury using isolated single cardiomyocytes, and show through images whether the cardiomyocyte mitochondrial membrane potential is maintained and ROS generation is suppressed during reperfusion by Erucin.
REPLY: We recognize that it would be very interesting; nevertheless, we have not the instrument to carry out a such evaluation. Indeed, we should isolate cardiomyocytes from heart and then submitted them to I/R injury following the effect of ERU on membrane potential as well as ROS production. To be honest it would be a very complex investigation. We please consider the previous our paper in which ERU showed protective effects in cardiomyoblasts (H9c2 cells) under H2O2-mediated oxidative stress (Testai et al., Biochem Pharmacol. 2023 doi:10.1016/j.bcp.2023.115728), suggesting that ERU can contain the ROS production deriving from a such type of environment.
- Since the information related to Erucin as an H2S donor was not shown in this experiment, please delete it from the abstract and briefly describe it in the introduction.
REPLY: we modified the abstract and describe the H2S-donor property of erucin in the introduction section, as you suggest.
Round 2
Reviewer 1 Report
Comments and Suggestions for Authors
The authors generally responded to my concerns. Regarding the assessment of oxidative stress markers, I suggest that the authors add this as a limitation of the study.
Author Response
The authors generally responded to my concerns. Regarding the assessment of oxidative stress markers, I suggest that the authors add this as a limitation of the study.
REPLY: we agree with you, therefore we added in the discussion this limitation.
Reviewer 2 Report
Comments and Suggestions for Authors
The authors faithfully responded to the reviewer's suggestions, and it is deemed sufficient for this paper to be published in the journal.
Author Response
The authors faithfully responded to the reviewer's suggestions, and it is deemed sufficient for this paper to be published in the journal.
REPLY:
Thanks for the comment.